# Deep Learning Reconstruction for DWIs by EPI and FASE Sequences for Head and Neck Tumors

**DOI:** 10.3390/cancers16091714

**Published:** 2024-04-28

**Authors:** Hirotaka Ikeda, Yoshiharu Ohno, Kaori Yamamoto, Kazuhiro Murayama, Masato Ikedo, Masao Yui, Yunosuke Kumazawa, Yurika Shimamura, Yui Takagi, Yuhei Nakagaki, Satomu Hanamatsu, Yuki Obama, Takahiro Ueda, Hiroyuki Nagata, Yoshiyuki Ozawa, Akiyoshi Iwase, Hiroshi Toyama

**Affiliations:** 1Department of Radiology, Fujita Health University School of Medicine, Toyoake 470-1192, Aichi, Japan; 2Department of Diagnostic Radiology, Fujita Health University School of Medicine, Toyoake 470-1192, Aichi, Japan; 3Joint Research Laboratory of Advanced Medical Imaging, Fujita Health University School of Medicine, Toyoake 470-1192, Aichi, Japan; 4Canon Medical Systems Corporation, Otawara 324-8550, Tochigi, Japan; 5Department of Radiology, Fujita Health University Hospital, Toyoake 470-1192, Aichi, Japan

**Keywords:** MRI, sequence, diffusion-weighted imaging, fast imaging, deep learning reconstruction

## Abstract

**Simple Summary:**

Diffusion-weighted images (DWI) obtained by echo-planar imaging (EPI) are frequently degraded by susceptibility artifacts from air-soft tissue interfaces at complicated structures in the paranasal sinus, oral cavity, pharynx, and larynx. It has been suggested that DWI obtained by fast advanced spin-echo (FASE) or reconstructed with deep learning reconstruction (DLR) could be useful for image quality improvements. The purpose of this investigation using in vitro and in vivo studies was to determine the influence of sequence difference and application of DLR for DWI on image quality, apparent diffusion coefficient (ADC) evaluation, and differentiation of malignant from benign head and neck tumors. In comparison with the EPI sequence, the FASE sequence and DLR can improve image quality and image distortion of DWIs without significantly influencing ADC measurements or the capability to differentiate malignant from benign head and neck tumors.

**Abstract:**

Background: Diffusion-weighted images (DWI) obtained by echo-planar imaging (EPI) are frequently degraded by susceptibility artifacts. It has been suggested that DWI obtained by fast advanced spin-echo (FASE) or reconstructed with deep learning reconstruction (DLR) could be useful for image quality improvements. The purpose of this investigation using in vitro and in vivo studies was to determine the influence of sequence difference and of DLR for DWI on image quality, apparent diffusion coefficient (ADC) evaluation, and differentiation of malignant from benign head and neck tumors. Methods: For the in vitro study, a DWI phantom was scanned by FASE and EPI sequences and reconstructed with and without DLR. Each ADC within the phantom for each DWI was then assessed and correlated for each measured ADC and standard value by Spearman’s rank correlation analysis. For the in vivo study, DWIs obtained by EPI and FASE sequences were also obtained for head and neck tumor patients. Signal-to-noise ratio (SNR) and ADC were then determined based on ROI measurements, while SNR of tumors and ADC were compared between all DWI data sets by means of Tukey’s Honest Significant Difference test. Results: For the in vitro study, all correlations between measured ADC and standard reference were significant and excellent (0.92 ≤ ρ ≤ 0.99, *p* < 0.0001). For the in vivo study, the SNR of FASE with DLR was significantly higher than that of FASE without DLR (*p* = 0.02), while ADC values for benign and malignant tumors showed significant differences between each sequence with and without DLR (*p* < 0.05). Conclusion: In comparison with EPI sequence, FASE sequence and DLR can improve image quality and distortion of DWIs without significantly influencing ADC measurements or differentiation capability of malignant from benign head and neck tumors.

## 1. Introduction

More than 90% of head and neck malignancies are accounted for by squamous cell carcinomas (SCC), followed by lymphomas [1]. While radiation therapy and surgical resection can lead to favorable results for early-stage head and neck SCC, the odds are less favorable for advanced stages. Therefore, early detection and diagnosis, as well as accurate staging for head and neck malignancies, are considered vital in routine clinical practice. 

During the last few decades, magnetic resonance imaging (MRI) has been continuously and widely used as a useful imaging technique for diagnosis, staging, therapeutic evaluation, and prediction of therapeutic effects for conservative therapies. Moreover, diffusion-weighted imaging (DWI) is being clinically used and has been considered one of the key sequences for the management of head and neck tumors since the early 2000s [2,3,4]. DWI is frequently obtained by means of echo-planar imaging (EPI) sequence and is considered a functional MRI technique for various clinical purposes, allowing for the quantification of the diffusion of water molecules in a tumor by measuring the apparent diffusion coefficient (ADC). However, one of the problems with using DWI for head and neck malignancies is the degradation of image quality, lesion conspicuity, and its effect on ADC measurements because of susceptibility artifacts from air-soft tissue interfaces at complicated structures in the paranasal sinus, oral cavity, pharynx, and larynx. Therefore, a few investigators have suggested during the past decade that, in comparison with EPI, spin-echo-based sequences, including fast advanced spin-echo (FASE), could be useful for DWI to overcome susceptibility artifacts and image distortion [5,6,7,8]. In addition, deep learning reconstruction (DLR) for image noise reduction on not only MRI but also CT was introduced in 2019 and has been in use since then [9,10,11,12,13]. However, no major studies have been published on the impact of DLR on quantitatively assessed DWI obtained by means of EPI and FASE sequences for head and neck malignancies. We hypothesized that DLR may be capable of improving image quality on DWI obtained by means of EPI as well as FASE sequences with little effect on the accuracy of differentiation of malignant from benign head and neck tumors. We also hypothesized that the FASE sequence can improve the image quality and accuracy of ADC measurements, regardless of whether or not DLR is used. The purpose of this study was thus to determine, by means of in vitro and in vivo studies, the influence of sequence difference and application of DLR for DWI on image quality, ADC evaluation, and differentiation of malignant from benign head and neck tumors. 

## 2. Materials and Methods

### 2.1. Protocol, Support, and Funding

This retrospective study was performed after obtaining institutional review board and written informed consents were waved from all subjects. Three of the authors are employees of Canon Medical Systems Corporation but did not have any control of the data in this study.

### 2.2. Quantitative Diffusion Phantom for In Vitro Study

As for the in vitro study, a diffusion phantom for quantitative study (High Precision Devices, Inc., Boulder, CO, USA), which was manufactured by the National Institute of Standards and Technology (NIST)/Radiological Society of North America (RSNA)-QIBA and recommended by the Quantitative Imaging Biomarker Alliance (QIBA), was used (Figure 1). This phantom contains 13 vials that have different concentrations of polyvinylpyrrolidone (PVP) in an aqueous solution from each other [14] to evaluate the ADC of the content of each vial. The phantom was designed for quantitative mapping of the isotropic Gaussian diffusion of water molecules and generating physiologically relevant ADC values [15]. The concentrations of PVP in the aqueous solutions were as follows: vials 1–3; 0%, vials 4–5; 10%, vials 6–7; 20%, vials 8–9; 30%, vials 10–11; 40%, and vials: 12–13; 50% [15]. The room between the vials inside the phantom was filled with an ice-water bath at 0 °C to reduce the thermal difference between scanner locations and time points in ADC measurements [14]. 

### 2.3. Subjects for In Vivo Study

From the beginning of October 2020 until the end of July 2021, 74 consecutive patients with various head and neck tumors (39 men and 35 women, mean age 60.4 years, age range 12–93 years) underwent head and neck MRI. These 74 patients were histopathologically or clinicoradiologically diagnosed with head and neck tumors who had not been treated yet and were originally included in this study. Exclusion criteria were: (1) pediatric patients who needed anesthesia or sedation during MR examinations, (2) MR examinations without DWIs obtained by EPI and FASE sequences, (3) tumors with a long-axis diameter of less than 10 mm, and (4) severe susceptibility artifacts resulting from dental metal. Of the initial 74 cases, 16 were excluded due to criterion (2) (n = 2), criterion (3) (n = 11), and criterion (4) (n = 3). The final study sample thus consisted of 58 patients (30 men, 28 women; median age, 66 years, age range, 12–93 years old). The flow chart for patient selection is shown in Figure 2. In addition, details of patient characteristics are shown in Table 1. 

### 2.4. MR Imaging

All MR examinations for both the in vitro and in vivo study were performed with a 3T MR system (Vantage Centurian, Canon Medical Systems Corporation, Otawara, Japan) using a multiple phased-array surface coil (Atlas SPEEDER Head/Neck, Canon Medical Systems). 

#### 2.4.1. MR Imaging for In Vitro Study

For the in vitro study, the phantom was obtained five times using DWI with EPI and FASE by means of a head and neck coil with parallel imaging capability (Atlas SPEEDER Head/Neck, Canon Medical Systems). The parameters of DWI with EPI were: Repetition Time (TR)/Echo Time (TE), 6800/78 ms; TI, 240 ms; number of slices, 40; slice thickness, 3 mm; Field of View (FOV), 250 × 240 mm^2^; acquisition matrix, 136 × 80; number of excitations, 2; flip angle (FA), 90/180; reduction factor (SPEEDER factor), 2.5; b-value 0, 800 s/mm^2^. The parameters of DWI with FASE were: TR/TE, 20,800/78 ms; number of slices, 40; slice thickness; 3 mm; FOV, 250 × 240 mm^2^; acquisition matrix, 136 × 80; NEX, 3; FA, 90/140; reduction factor (SPEEDER factor), 2.5; b value 0, 800 s/mm^2^. Each data set of DWI with EPI and FASE was then reconstructed with and without DLR (Advanced intelligent Clear IQ Engine: AiCE, Canon Medical Systems). Details of DLR have been specified in previously published studies [12,13]. 

#### 2.4.2. MR Imaging for In Vivo Study

As for the in vivo study, each subject underwent head and neck MR imaging with axial T2-weighted fast spin-echo (FSE) imaging and axial DWIs with single-shot EPI and FASE sequences using the same head and neck coil. Details of the scan protocols in this study are shown in Table 2. 

For all patients, T2-weighted image (T2WI) with FSE was firstly obtained by compressed sensing with parallel imaging technique (Compressed SPEEDER, Canon Medical Systems) with the following parameters: repetition time (TR), 4800 ms; echo time (TE), 93.5 ms; flip angle (FA), 90/160 deg; acquisition matrix, 352 × 352; reconstruction matrix, 704 × 704; slice thickness, 3 mm; slice gap, 0 mm; number of slices, 40; field of view (FOV), 200 × 200 mm^2^; number of excitations (NEX), 1; reduction factor (Compressed Speeder factor), 3. Details of compressed sensing with parallel imaging techniques have been written in previously published articles [12,16]. Secondly, DWIs with EPI sequences (DWI_EPI_) were obtained by short inversion-time recovery (STIR) technique with the following parameters: TR, 6800 ms; TE, 78 ms; inversion time (TI), 240 ms; FA, 90/180 degrees; acquisition matrix, 136 × 80; reconstruction matrix, 272 × 260; slice thickness, 3 mm; slice gap, 0mm; number of slices, 40; FOV, 250 × 240 mm^2^; NEX, 2; reduction factor (SPEEDER factor), 2.5; b value, 0 and 800 s/mm^2^. Thirdly, DWIs with the FASE sequence (DWI_FASE_) were obtained with STIR with short TI technique using the following parameters: TR, 20,800 ms; TE, 78 ms; TI, 240 ms; FA, 90/140 degrees; acquisition matrix, 136 × 80; reconstruction matrix, 272 × 260; slice thickness, 3 mm; slice gap, 0 mm; number of slices, 40; FOV, 250 × 240 mm^2^; NEX, 3; reduction factor (SPEEDER factor), 2.5; b value, 0 and 800 s/mm^2^. Finally, each data set of DWI with EPI and FASE was reconstructed with and without DLR. 

### 2.5. Image Analysis

#### 2.5.1. Image Analysis for In Vitro Study

Signal intensity data obtained from each voxel on DWIs were fitted to a mono-exponential model to calculate the ADC by using the following built-in Tensor application (System software version 6.0: Canon Medical Systems): S(b) = S_0_ exp(−bADC)(1)
where S(b) and S_0_ are the signal intensities with and without MPG pulse, respectively, and the quantity b equals the b-value (s/mm^2^). ADC for each phantom was measured by a board-certified head and neck radiologist (H.I.) with 15 years of experience. Circular regions of interest (ROIs) 10 mm in diameter were placed on the center slice and on another two slices 1.2 cm on either side of the center slice on each phantom, after which the mean ADC value within each phantom was calculated.

#### 2.5.2. Image Analysis for In Vivo Study

Image quality assessment was performed by using System software (version 6.0, Canon Medical Systems). A board-certified head and neck radiologist (H.I.) with 15 years of experience performed region of interest (ROI) measurements. Free-hand ROIs were set over each head and neck tumor on DWI_FASE_ and DWI_EPI_ with and without DLR, then the tumor’s signal-to-noise ratio (SNR) and apparent diffusion coefficient (ADC) were measured. SNRs were calculated with the following formula based on previously published studies [16,17,18]: SNR = SI_tumor_/SD_tumor_(2)
where SI_tumor_ and SD_tumor_ are the signal intensity and standard deviation of the tumor in the ROI placed on the tumor.

To determine the difference in deformation among all DWIs, the margin of each tumor was traced by free-hand ROI on the same level for all sequences, and then the tumor’s area on each sequence was calculated. The deformation ratio (DR) was defined as the difference in the free-hand ROI area between each DWI and T2WI divided by the ROI area on T2WI. The formula of the calculation is as follows:DR = |Area_T2WI_ − Area_DWI_|/Area_T2WI_(3)
where Area_T2WI_ and Area_DWI_ are the tumor’s area on T2WI and each DWI sequence.

Another freehand ROI was placed as a reference over the spinal cord on the same level as the tumor on DWI_FASE_ and DWI_EPI_ with and without DLR, as in previous studies [5,19]. SNR and DR of the spinal cord on each DWI were also calculated using the same formula as the tumor shown above (Formulas (2) and (3)). Furthermore, to compare the difference in SNR and DR of tumors according to tumor location, all tumors were categorized into 4 groups according to their location in head and neck: (1) pharynx and larynx, (2) oral cavity and mandibular, where tumors often interfere with air, (3) salivary glands and (4) others, where tumors do not usually interfere with air. 

### 2.6. Statistical Analysis

#### 2.6.1. In Vitro Study 

Statistical Analysis Was Performed Using JMP Software (version 14, SAS Institute Inc., Cary, NC, USA).

To compare the measurement accuracy for ADC among DWIs obtained by EPI and FASE sequences and reconstructed with and without DLR, correlations between ADC values measured with each method and the standard reference among all phantoms were evaluated by Spearman’s rank correlation analysis. Measurement errors for ADC evaluation were then compared by means of the Wilcoxon signed-rank test among DWIs obtained by EPI and FASE sequences and reconstructed with and without DLR.

#### 2.6.2. In Vivo Study

JMP software (version 14, SAS Institute Inc., Cary, NC, USA) was used for statistical analysis.

Tukey’s HSD test was performed to compare image acquisition and reconstruction times among all DWIs.

For a quantitative comparison of image quality among DWIs obtained by EPI and FASE sequences and reconstructed with and without DLR, SNR, and DR of overall tumors and of the spinal cord were compared among all methods by one-way ANOVA followed by Tukey’s Honest Significant Difference (HSD) test. Moreover, the SNR and DR of a tumor in a given group were compared among all methods by using the same statistical analysis to determine the relationship between image quality and tumor location.

A Student *t*-test was used to compare the ADC values of benign and malignant tumors for each DWI. To assess the influence of sequence difference and DLR on ADC measurements, ADC values of benign and malignant tumors were also statistically compared among all DWI methods by means of Tukey’s HSD test. 

To determine the capability of each method to differentiate malignant from benign tumors, a receiver operating characteristic (ROC)-based positive test was performed. Because Warthin tumor, adenoid hypertrophy and benign lymphadenopathy are known to show low ADC values though they are benign [20,21,22,23], additional ROC-based positive tests were also performed after patients with these etiologies had been excluded. 

Finally, by applying each feasible threshold value, sensitivity, specificity, and accuracy were compared by means of McNemar’s test among all DWI methods for all head and neck tumors except for Warthin tumor, adenoid hypertrophy, and benign lymphadenopathy. 

A *p*-value of less than 0.05 was considered significant for all statistical analyses. 

## 3. Results

### 3.1. In Vitro Study

Correlations between measured ADC and standard reference for all DWI methods are shown in Table 3. Correlations between measured ADC and standard reference for all methods were significant and excellent (0.92 ≤ ρ ≤ 0.99, *p* < 0.0001). 

A comparison of measurement errors for ADC evaluation between EPI with and without DLR and between FASE with and without DLR is shown in Table 4. Measurement errors for FASE with DLR were significantly larger than those of FASE without DLR at all fantom concentrations (*p* < 0.001). 

### 3.2. In Vivo Study

The final group included 58 patients (30 men, 28 women; mean age, 60.6 ± 19.0 years; age range, 12–93 years). There were 15 patients in (1) the pharynx and larynx group, 12 patients in (2) the oral cavity group, 15 patients in (3) the salivary glands group, and 16 patients in (4) the group of ‘others’. 

A representative case is shown in Figure 3. 

A comparison of results for image acquisition and reconstruction times among DWIs_EPI_ with and without DLR and DWIs_FASE_ with and without DLR is shown in Figure 4. Mean image acquisition and reconstruction times for DWIs with DLR (DWI_EPI_: 223.1 ± 9.6 s, DWI_FASE_: 437.0 ± 17.3 s) were significantly longer than for those without DLR (DWI_EPI_: 206.8 ± 8.9 s, *p* < 0.0001; DWI_FASE_: 420.8 ± 16.8 s, *p* < 0.0001). In addition, mean image acquisition and reconstruction times for DWI_FASE_ with and without DLR were significantly longer than for those DWI_EPI_ with and without DLR (*p* < 0.0001). 

Comparisons of SNR of overall tumors, the spinal cord and each tumor group for all DWI methods are shown in Table 5. SNRs of FASE with DLR (5.6 ± 2.4) were significantly higher than those of FASE without DLR in overall tumors (4.3 ± 1.6, *p =* 0.02). 

Comparisons of DR of overall tumors, the spinal cord, and each tumor group for each sequence with and without DLR and all DWIs are shown in Table 6. For overall tumors, DRs of FASE with and without DLR (with DLR: 0.13 ± 0.17, without DLR: 0.19 ± 0.13) were significantly lower than those of EPI with and without DLR (with DLR: 0.37 ± 0.33, vs. FASE with DLR, *p* < 0.0001, vs. FASE without DLR, *p* = 0.001; without DLR: 0.36 ± 0.34, vs. FASE with DLR, *p* < 0.0001, vs. FASE without DLR, *p* = 0.003). For the spinal cord, DR of FASE with DLR (0.13 ± 0.13) was significantly lower than that of EPI with and without DLR (with DLR: 0.34 ± 0.30, *p* < 0.0001; without DLR: 0.31 ± 0.30, *p =* 0.0003). Moreover, the DR of FASE with DLR (0.10 ± 0.10) was significantly lower than that of EPI with and without DLR (with DLR: 0.33 ± 0.92, *p* = 0.03; without DLR: 0.32 ± 0.30, *p* = 0.04) for the pharynx and larynx group. In addition, the DR of FASE with DLR (0.06 ± 0.05) was significantly lower than that of EPI with and without DLR (with DLR: 0.47 ± 0.31, *p* = 0.001; without DLR: 0.40 ± 0.37, *p* = 0.009) for the oral cavity and mandibular group. 

Comparisons of ADCs for malignant and benign tumors between each DWI with and without DLR are shown in Table 7. ADC values for benign and malignant tumors showed significant differences between each sequence with and without DLR (*p* < 0.05).

Feasible threshold values and diagnostic performance of all DWI methods for all patients are shown in Table 8. The application of each threshold value for differentiating malignant tumors from benign tumors showed no significant differences in sensitivity, specificity, or accuracy for all DWI methods (*p* > 0.05). Feasible threshold values and diagnostic performance for all DWI methods for all head and neck tumors except for Warthin tumor, adenoid hypertrophy, and benign lymphadenopathy are shown in Table 9. The application of each threshold value for differentiating malignant tumors from benign tumors showed no significant differences in sensitivity, specificity, or accuracy for all DWI methods (*p* > 0.05). 

## 4. Discussion

Our in vitro and in vivo study results demonstrate that DWI_FASE_ with DLR delivered better quantitative image quality without little influence on ADC measurements than did DWI_FASE_ without DLR on a 3T MR system. Moreover, the in vivo study findings showed that, in comparison with DWI_EPI_, using DWI_FASE_ could reduce image distortions in head and neck tumors, especially those in the pharynx, larynx, oral cavity, and mandibular groups, regardless of whether DLR was used or not. However, there were no statistically significant differences between DWI_EPI_ and DWI_FASE_ for distinguishing malignant from benign head and neck tumors. To the best of our knowledge, this study was the first to use DLR for DWIs obtained by FASE and EPI sequences in in vivo and in vitro studies and determine its effect on image quality, ADC measurements, and differentiation between malignant and benign head and neck tumors. 

As for the in vitro study, DWIs obtained with any of the methods showed significant and excellent correlations between measured ADC and the standard reference. However, DWI_FASE_ was also considered less tolerant for ADC measurements than DWI_EPI_, and DLR had a greater effect on ADC measurements when DWI_FASE_ was used. 

In addition to the in vitro study, our in vivo study demonstrated that image acquisition and reconstruction times for DWI_FASE_ with or without DLR were significantly longer than those for DWI_EPI_ with or without DLR and that DLR could significantly improve SNR on DWI obtained by FASE sequence. These findings were comparatively compatible with those of previous studies [10,12,13,24,25,26,27,28,29]. Moreover, our study demonstrated that the overall distortion rates of DWI_FASE_ with or without DLR were significantly smaller than those of DWI_EPI_ with or without DLR. In addition, distortion ratios of DWI_FASE_ with DLR showed significant improvements over those of DWI_EPI_ with and without DLR for lesions in the “spinal cord”, “pharynx and larynx” and “oral cavity and mandibular” groups. The findings of the in vivo study indicate that the FASE sequence is potentially superior for head and neck DWIs to the EPI sequence in terms of improvements in image quality and ADC measurement. Findings of previous studies, which evaluated head and neck DWIs obtained with spin-echo (SE)-based sequences such as turbo SE or FASE sequence, suggest that DWIs thus obtained are more tolerant than DWIs obtained with EPI for various anatomical areas [5,6,7,8,9,10,30,31,32]. Considering these previous findings and our results, it can be easily assumed that DWI would be better than EPI for obtaining FASE sequences for head and neck tumors reconstructed with DLR in routine clinical practice. 

The comparison of ADC values for benign and malignant tumors showed significant differences between each DWI reconstructed with and without DLR, although no significant differences in diagnostic performance were observed among all DWI data sets. This finding is compatible with those previously published for DWIs for prostatic cancer in a study [13]. It, therefore, seems preferable to obtain DWIs by using the FASE rather than the EPI sequence to improve image distortion, and it might be better to use DLR for image quality improvement for either sequence in head and neck tumor patients. 

There are a few limitations to this study. First, the study population was small, and underlying pathological and clinical situations were varied so that no significant improvements in diagnostic performance could be observed. Large prospective cohort studies are thus warranted to demonstrate the actual clinical significance of DWI_FASE_ or DLR in this setting. Second, the conventional parallel imaging technique was used for DWI_FASE_ and DWI_EPI_ acquisitions. However, recently clinically applied compressed sensing (CS) with and without parallel imaging or FASE acquisition with multiple k-space data acquisition using various time reduction techniques have been introduced as useful for acquisition time reduction [12,16,33] but have not been used or tested in this study. Moreover, DLR in this study was provided by a single vendor, so DLRs from other vendors could not be used. It would, therefore, be better to use DWI_FASE_ with other acquisition techniques and other DLR methods in future studies to demonstrate the actual clinical significance of DWI_FASE_ in this setting. Third, although free-hand ROI placement was carefully performed by an experienced head and neck radiologist, there were potential errors and variability outlining the head and neck tumors. ROI placement by blinded multiple radiologists could decrease the potential error and variability of ROI measurement. 

## 5. Conclusions

In conclusion, in comparison with the EPI sequence for the 3T MR system, the FASE sequence and DLR can improve image quality and image distortion of DWIs without significantly influencing ADC measurements in the in vivo or in vitro study or the capability to differentiate malignant from benign head and neck tumors. 

## Figures and Tables

**Figure 1 cancers-16-01714-f001:**
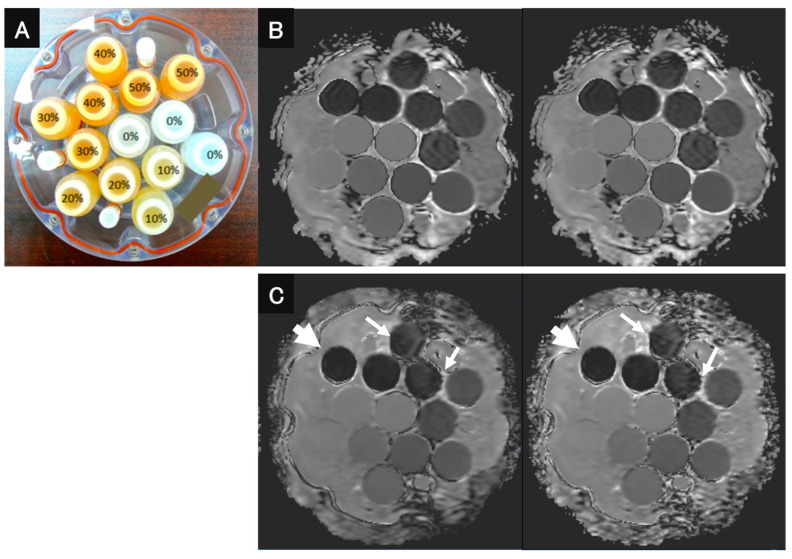
QIBA recommended that DWI phantom and DWI images be obtained with both sequences and reconstructed with and without DLR. (**A**) QIBA recommended DWI phantom using 13 vials with different concentrations of PVP (polyvinylpyrrolidone). (**B**) (L to R: DWI_EPI_ reconstructed with and without DWI) DWI_EPI_ with DLR was not significantly different from DWI_EPI_ without DLR. (**C**) (L to R: DWI_FASE_ reconstructed with and without DWI) Image noise for DWI_FASE_ with DLR was less than for DWI_FASE_ without DLR (arrows). In addition, the image distortion of DWI_FASE_s with and without DLR showed less image distortion than DWI_EPI_s with and without DLR (large arrows).

**Figure 2 cancers-16-01714-f002:**
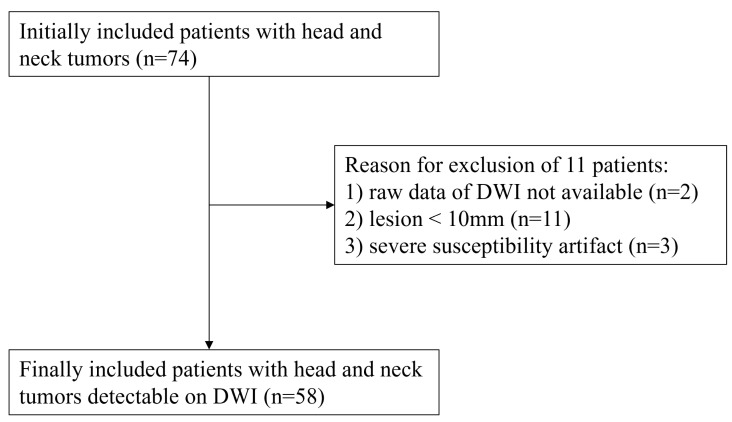
Patient selection flow chart. The flow chart shows the patient selection from the original 74 patients to the final 58 patients.

**Figure 3 cancers-16-01714-f003:**
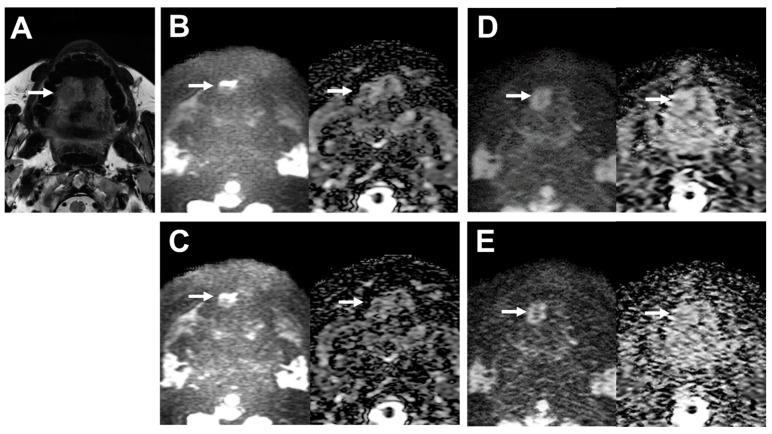
39-year-old male patient with oral cavity squamous cell carcinoma. (**A**) T2WI clearly depicts the lobulated shape of oral cavity cancer at the right side of the tongue (arrow). (**B**) (L to R, original DWI image and ADC map) Image of the tumor on DWI obtained by EPI with DLR shows deformation (arrows). SNR of the tumor was 4.43, and the deformation ratio (DR) compared with T2WI was 0.60. (**C**) (L to R, original DWI image and ADC map) The image of the tumor on DWI obtained by EPI without DLR is also deformed (arrows). SNR of the tumor was 4.58, and the deformation ratio (DR) compared with T2WI was 0.60. (**D**) (L to R, original DWI image and ADC map) The image of the tumor on DWI obtained by FASE with DLR is less deformed (arrows). SNR of the tumor was 7.08, and the deformation ratio (DR) compared with T2WI was 0.01. (**E**) (L to R, original DWI image and ADC map) The image of the tumor on DWI obtained by FASE without DLR is deformed (arrows). SNR of the tumor was 4.47, and the deformation ratio (DR) compared with T2WI was 0.21.

**Figure 4 cancers-16-01714-f004:**
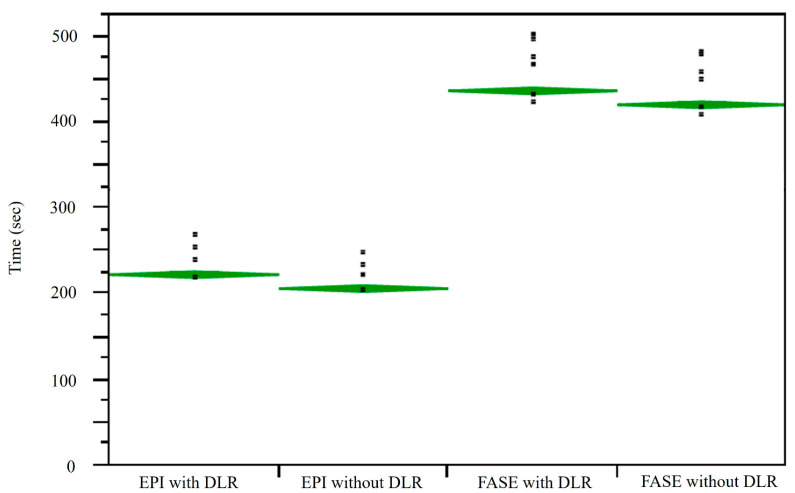
Comparison of image acquisition and reconstruction times for DWI_EPI_ with and without DLR and for DWI_FASE_ with and without DLR. L to R: EPI with and without DLR, FASE with and without DLR. Mean image acquisition and reconstruction times for DWI_FASE_ with and without DLR were significantly longer than those for DWI_EPI_ with and without DLR (*p* < 0.0001). Mean image acquisition and reconstruction times for DWI_EPI_ and DWI_FASE_ with DLR were significantly longer than those for DWI_EPI_ and DWI_FASE_ without DLR (*p* < 0.0001).

**Table 1 cancers-16-01714-t001:** Patient characteristics.

**Gender**	**Male**	30	
	**Female**	28	
**Age**	**Male**	Mean: 62.8, range: 12 to 89 years old	
	**Female**	Mean: 58.2, range: 17 to 93 years old	
**Histology**	**Malignant tumors**	Nasopharyngeal cancer	1
		Oropharyngeal cancer	3
		Hypopharyngeal cancer	4
		Oral cancer	7
		Laryngeal cancer	5
		Maxillary sinus cancer	4
		Thyroid cancer	1
		Follicular cell lymphoma	1
	**Benign lesions**	Venous malformation	2
		Nasopharyngeal adenoid hypertrophy	1
		Pleomorphic adenoma of parotid gland	7
		Warthin tumor of parotid gland	4
		Parotid gland	1
		Sjogren syndrome	2
		Sarcoidosis	1
		Ranula	1
		Thyroglossal duct cyst	1
		Second brachial cleft cyst	1
		Benign follicular tumor thyroid	2
		Benign neurogenic tumor	4
		Calcifying epithelial odontogenic tumor	1
		Odontogenic keratocyst	2
		Benign lymphadenopathy	1
		Unknown benign mandibular lesion	1

**Table 2 cancers-16-01714-t002:** Details of scan protocols for in vivo study.

Protocol	T2WI	DWI by EPI	DWI by FASE
Sequence	Multi-slice FSE	Single-shot EPI	Single-shot FASE
Fat suppression	N/A	STIR	STIR
Repetition time (TR: ms)	4500–5300	6800–7964	26,628–30,000
Echo time (TE: ms)	93.5	78	78
Inversion Time (TI: ms)	N/A	240	240
Echo train length (ETL)	19	-	-
Echo spacing (ETS: ms)	8.5	0.52	0.8
Acquisition matrix	352 × 352	136 × 80	136 × 80
Reconstruction matrix	704 × 704	272 × 260	272 × 260
Reconstruction matrix size (mm)	0.3 × 0.3	0.9 × 0.9	0.9 × 0.9
Field of view (FOV: mm)	200 × 200	250 × 240	250 × 240
Number of phase wraps	1.8	N/A	N/A
Slice thickness (mm)	3	3	3
Number of Slices	40–47	40–47	40–47
Number of excitations (NEX)	1	2	3
Flip angle (degree)	90/160	90/180	90/140
Phase encoding direction	RL	AP	AP
Imaging plane	Axial	Axial	Axial
Acceleration method	Compressed SPEEDER	SPEEDER	SPEEDER
Reduction factor	3	2.5	2.5
b-value (s/mm^2^)	-	0.800	0.800
Acquisition time (s)	109 (108–162)	206 (204–249)	420 (408–481)

T2WI: T2-weighted imaging, EPI: echo planar imaging, FASE: Fast advanced spin-echo, N/A: not applicable, STIR: short inversion time (TI) inversion recovery.

**Table 3 cancers-16-01714-t003:** Correlations for ADC between standard reference and measured ADC for all DWI methods.

	EPI without DLR		EPI with DLR		FASE without DLR		FASE with DLR	
	ρ	*p*	ρ	*p*	ρ	*p*	ρ	*p*
EPI without DLR	N/A	N/A	0.99	<0.0001	0.96	<0.0001	0.92	<0.0001
EPI with DLR	–	–	N/A	N/A	0.96	<0.0001	0.93	<0.0001
FASE without DLR	–	–	–	–	N/A	N/A	0.92	<0.0001
FASE with DLR	–	–	–	–	–	–	N/A	N/A

N/A: not applicable.

**Table 4 cancers-16-01714-t004:** Comparison of measurement errors in ADCs for EPI and FASE with and without DLR.

Concentration of Phantom [%]	Standard Reference [×10^−3^ mm^2^/s]	Mean Difference ± SD					
		EPI without DLR	EPI with DLR	*p* Value	FASE without DLR	FASE with DLR	*p* Value
0	1.127	0.011 ± 0.034	0.012 ± 0.034	0.7962	0.033 ± 0.077	0.132 ± 0.056	<0.0001
10	0.843	0.000 ± 0.045	0.000 ± 0.044	0.9941	0.052 ± 0.098	0.163 ± 0.077	<0.0001
20	0.607	0.015 ± 0.028	0.016 ± 0.029	0.8303	0.042 ± 0.104	0.182 ± 0.074	<0.0001
30	0.403	0.035 ± 0.083	0.042 ± 0.084	0.6520	0.031 ± 0.154	0.252 ± 0.103	<0.0001
40	0.248	0.113 ± 0.084	0.105 ± 0.080	0.6414	0.070 ± 0.137	0.057 ± 0.158	0.0005
50	0.128	0.031 ± 0.034	0.027 ± 0.033	0.5493	0.013 ± 0.047	0.037 ± 0.052	0.0002

**Table 5 cancers-16-01714-t005:** Comparison of signal-to-noise ratio (SNR) for tumors and spinal cord.

	Number of Cases	EPI without DLR	EPI with DLR	FASE without DLR	FASE with DLR
Overall tumors	58	5.1 ± 2.6	5.4 ± 2.8	4.3 ± 1.6	5.6 ± 2.4 *
Spinal cord	58	8.7 ± 6.2	8.7 ± 5.7	7.0 ± 3.1	9.3 ± 3.8
Pharynx and larynx	15	5.6 ± 3.0	5.9 ± 3.1	5.1 ± 2.3	7.0 ± 3.1
Oral cavity and mandibular	12	4.8 ± 2.7	5.6 ± 3.0	3.6 ± 1.2	5.0 ± 1.9
Salivary glands	15	3.9 ± 1.5	4.0 ± 1.7	4.0 ± 1.2	4.3 ± 1.3
Others	16	5.3 ± 2.2	6.2 ± 3.0	4.4 ± 1.1	5.9 ± 2.3

* Significantly higher than FASE without DLR (*p* < 0.05).

**Table 6 cancers-16-01714-t006:** Comparison of deformation ratio (DR) for tumors and spinal cord.

	Number of Cases	EPI without DLR	EPI with DLR	FASE without DLR	FASE with DLR
Overall tumors	58	0.36 ± 0.34	0.37 ± 0.33	0.19 ± 0.13 *^,^**	0.13 ± 0.17 *^,^**
Spinal cord	58	0.31 ± 0.30	0.34 ± 0.30	0.23 ± 0.14	0.13 ± 0.13 *^,^**
Pharynx and larynx	15	0.32 ± 0.30	0.33 ± 0.92	0.14 ± 0.10	0.10 ± 0.10 *^,^**
Oral cavity and mandibular	12	0.40 ± 0.37	0.47 ± 0.31	0.22 ± 0.18	0.06 ± 0.05 *^,^**
Salivary glands	15	0.31 ± 0.45	0.35 ± 0.48	0.17 ± 0.08	0.12 ± 0.12
Others	16	0.42 ± 0.28	0.36 ± 0.24	0.22 ± 0.19	0.22 ± 0.26

* Significantly lower than EPI without DLR (*p* < 0.05). ** Significantly lower than EPI with DLR (*p* < 0.05).

**Table 7 cancers-16-01714-t007:** A comparison of ADCs for malignant and benign tumors was obtained with each DWI method, with and without DLR, and with all methods.

Method	ADC [×10^−3^ mm^2^/s] (Mean ± SD)	
	Benign	Malignant
EPI without DLR	1.63 ± 0.53	1.10 ± 0.17 *
EPI with DLR	1.69 ± 0.57	1.16 ± 0.21 *
FASE without DLR	1.82 ± 0.56	1.22 ± 0.28 *
FASE with DLR	1.82 ± 0.56	1.21 ± 0.29 *

* Significantly different from ADC value for benign tumor (*p* < 0.05). There were no significant differences in ADCs for benign and malignant tumors obtained with all methods (*p* > 0.05).

**Table 8 cancers-16-01714-t008:** Results of comparison of diagnostic performance of DWI methods with and without DLR for all tumors.

	Feasible Threshold Value (×10^−3^mm^2^/s)	SE (%)	SP (%)	PPV (%)	NPV (%)	AC (%)
EPI with DLR	1.3	84.6 (22/26)	81.3 (26/32)	78.6 (22/28)	86.7 (26/30)	82.8 (48/58)
EPI without DLR	1.2	76.9 (20/26)	84.4 (27/32)	80.0 (20/25)	81.8 (27/33)	81.0 (47/58)
FASE with DLR	1.4	80.8 (21/26)	81.3 (26/32)	77.8 (21/27)	83.9 (26/31)	81.0 (47/58)
FASE without DLR	1.5	88.5 (23/26)	78.1 (25/32)	76.7 (23/30)	89.3 (25/28)	82.8 (48/58)

SE: sensitivity, SP: specificity, PPV: positive predictive value, NPV: negative predictive value, AC: accuracy.

**Table 9 cancers-16-01714-t009:** Comparison of Diagnostic Performance among all DWIs for all head and neck tumors except for Warthin tumor, adenoid hypertrophy, and benign lymphadenopathy.

	Feasible Threshold Value (×10^−3^mm^2^/s)	SE (%)	SP (%)	PPV (%)	NPV (%)	AC (%)
EPI with DLR	1.4	92.3 (24/26)	76.9 (20/26)	80.0 (24/30)	90.9 (20/22)	84.6 (44/52)
EPI without DLR	1.3	88.5 (23/26)	84.6 (22/26)	85.2 (23/27)	88.0 (22/25)	86.5 (45/52)
FASE with DLR	1.5	88.5 (23/26)	80.8 (21/26)	82.1 (23/28)	87.5 (21/24)	84.6 (44/52)
FASE without DLR	1.5	88.5 (23/26)	88.5 (23/26)	88.5 (23/26)	88.5 (23/26)	88.5 (46/52)

SE: sensitivity, SP: specificity, PPV: positive predictive value, NPV: negative predictive value, AC: accuracy.

## Data Availability

The datasets presented in this study are available from the corresponding author upon reasonable request.

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
