# Peer review of "Deep Learning Reconstruction for DWIs by EPI and FASE Sequences for Head and Neck Tumors"

_cancers, 2024, doi:10.3390/cancers16091714_

Round 1
Reviewer 1 Report
Comments and Suggestions for Authors
Please see some of the major concerns listed below:
MATERIALS AND METHODS
Although it's clear why certain patients, such as those with patent PFO, were excluded, the rationale behind excluding patients with lesions close to bony structures is not fully explained. How does this exclusion criterion impact the generalizability of the study's findings, especially considering the potential prevalence of such lesions in clinical practice?
The method of rigid image registration and manual outlining of structures raises questions about potential sources of error and variability. How were these sources of error minimized or accounted for in the study?
Reviewer 2 Report
Comments and Suggestions for Authors
Manuscript title: Deep Learning Reconstruction for DWIs by EPI and FASE Sequences for Head and Neck Tumors
1. The authors provide insights on qualitative and quantitative effect of deep learning reconstruction in diffusion-weighted images for neck tumor magnetic resonance imaging.
2. The primary study strength is detailed study methodology and high relevance to the field of quantitative MRI. Weaknesses include small and heterogeneous sample, absent information on used statistical software.
3. The conclusions are consistent with the evidence provided.
4. Figure 1 does not have arrows (contrary to the caption).
5. Data availability and ethics statements are inadequate.
Round 2
Reviewer 1 Report
Comments and Suggestions for Authors
The authors have satisfactorily addressed my concerns.